# Extracellular Vesicles and Acute Kidney Injury: Potential Therapeutic Avenue for Renal Repair and Regeneration

**DOI:** 10.3390/ijms23073792

**Published:** 2022-03-30

**Authors:** Maja Kosanović, Bojana Milutinovic, Sofija Glamočlija, Ingrid Mena Morlans, Alberto Ortiz, Milica Bozic

**Affiliations:** 1Institute for the Application of Nuclear Energy, INEP, University of Belgrade, 11080 Belgrade, Serbia; maja@inep.co.rs (M.K.); sofija.glamoclija@inep.co.rs (S.G.); 2Department of Neurosurgery, MD Anderson Cancer Center, University of Texas, Houston, TX 77030, USA; bmilutinovic@mdanderson.org; 3Vascular and Renal Translational Research Group, Biomedical Research Institute of Lleida Dr. Pifarré Foundation (IRBLleida), 25196 Lleida, Spain; imena@irblleida.cat; 4Department of Nephrology and Hypertension, IIS-Fundación Jiménez Díaz, Universidad Autónoma Madrid, 28040 Madrid, Spain; aortiz@fjd.es

**Keywords:** extracellular vesicles, exosomes, microvesicles, cellular communication, acute kidney injury, therapeutic agents, renoprotection, mesenchymal stem cells, cell-free therapeutic

## Abstract

Acute kidney injury (AKI) is a sudden decline of renal function and represents a global clinical problem due to an elevated morbidity and mortality. Despite many efforts, currently there are no treatments to halt this devastating condition. Extracellular vesicles (EVs) are nanoparticles secreted by various cell types in both physiological and pathological conditions. EVs can arise from distinct parts of the kidney and can mediate intercellular communication between various cell types along the nephron. Besides their potential as diagnostic tools, EVs have been proposed as powerful new tools for regenerative medicine and have been broadly studied as therapeutic mediators in different models of experimental AKI. In this review, we present an overview of the basic features and biological relevance of EVs, with an emphasis on their functional role in cell-to-cell communication in the kidney. We explore versatile roles of EVs in crucial pathophysiological mechanisms contributing to AKI and give a detailed description of the renoprotective effects of EVs from different origins in AKI. Finally, we explain known mechanisms of action of EVs in AKI and provide an outlook on the potential clinical translation of EVs in the setting of AKI.

## 1. Introduction

Acute kidney injury (AKI) is a clinical syndrome characterized by an abrupt worsening of renal function and represents a global clinical health issue associated with high short-term and increased long-term mortality [1]. AKI can be triggered by a variety of insults such as nephrotoxic medications, environmental contaminants, ischemia, systemic inflammation, nephritis, and urinary tract obstruction [1]. While discovering effective therapies for AKI is a continuous quest, currently only supportive therapy is available. Owing to the substantial complexity of the kidney structure and multifactorial pathophysiology of renal injury, regenerative medicine for kidney diseases remains unattainable. However, stem cell technology has shown great promise and over the past decades contemporary stem cell therapies have been applied in various preclinical models and clinical trials [2,3,4]. More importantly, extracellular vesicles (EVs) derived from stem cells have raised great interest as a new therapeutic option for accelerating recovery after different types of injury [5].

EVs are heterogeneous nanoparticles released by multiple cell types. EVs are surrounded by a bilayer membrane and contain a biomolecular cargo that reflects the physiological state of the parental cells from which vesicles originated [6]. Having a critical role in long-distance communication between cells, EVs represent important players in various physiological and pathological conditions, including kidney disease [7,8]. Moreover, due to their unique characteristics, EVs have great potential in the design of novel therapeutic strategies for different diseases [5]. Use of stem cell therapy to recover damaged cells is a generally known biomedical approach in the field of regenerative medicine [2]. However, EVs have great potential in replacing cellular therapies, overcoming challenges faced by the clinical application of stem cell therapy [5]. Hence, EVs secreted by mesenchymal stem cells have arisen as a powerful cell-free therapy for numerous disease conditions, AKI being one of them.

The current review outlines the functional roles of EVs in intercellular communication along the nephron and describes their impact on crucial pathophysiological mechanisms contributing to kidney injury. We summarize the latest findings on the renoprotective effects of EVs from diverse sources in experimental models of AKI and depict cellular and signaling mechanisms underlying these effects on renal tissue. Finally, we discuss the potential clinical translation of EVs in the context of AKI.

## 2. Acute Kidney Injury

AKI is a severe clinical syndrome defined by an abrupt decrease of renal function [1] frequently resulting from the injury of cells within the nephron. Being highly prevalent in hospitalized patients, AKI represents a significant global clinical problem, as it is associated with an elevated morbidity and mortality. A systematic analysis of 312 cohorts demonstrated that approximately 22% of adults and 34% of children worldwide experience AKI during a period of hospitalization [9]. In spite of significant progress in prophylactic strategies and supportive care, AKI remains associated with an increased morbidity and mortality, where mortality rates for intensive care unit patients can surpass 50% and the increased risk of death may persist for over a year [1]. The high mortality associated with AKI was most recently confirmed during the COVID-19 pandemic [10].

The pathogenesis of AKI is very complex and not entirely understood. Different triggering events may recruit related pathways of injury. During AKI, renal proximal tubular epithelial cells (RPTECs) are the principal targets of injury due to their large metabolic activity and high energy demands. Renal tubules perform essential functions of reabsorption and secretion to form urine and also secrete bioactive molecules, such as active vitamin D (1,25 (OH)2 vitamin D), and the antiaging, anti-inflammatory, and antifibrotic protein Klotho; thus, damage to renal tubules is a key contributor to AKI and to its consequences [11]. The pathophysiology of AKI encompasses renal tubular epithelial cell damage, apoptosis, oxidative stress, inflammation, and vascular dysfunction [12]. Depending on the severity of injury, AKI can lead to regeneration of the injured tissue or progress towards chronic kidney disease (CKD), which is known as AKI-to-CKD transition [13].

RPTECs of the healthy adult kidney are terminally differentiated epithelial cells that do not undergo proliferation; very few cells in a healthy kidney undergo cell division [14]. However, in response to injury, the surviving RPTECs undergo dedifferentiation, adopt proliferative phenotypes, and repopulate damaged tubules, thus repairing kidney damage [15]. This process of tubule regeneration after AKI encompasses activation of different signal transduction pathways in injured and regenerating epithelial cells, and is accompanied by the production and secretion of multiple cytokines, growth factors, and proinflammatory and profibrotic molecules [13]. In addition to RPTECs, endothelial or interstitial cells and recruited inflammatory cells can also be the source of these molecules [13]. Secreted bioactive molecules can exert autocrine functions, thus playing a role in epithelial dedifferentiation, migration, and the proliferation necessary for regeneration of renal tubules, while their paracrine functions are reflected in their chemotactic actions on circulating leukocytes [13]. Normal repair processes can result in full restoration of tubular epithelial structure and function (Figure 1A). However, if injury persists or repair processes fail, defective regeneration results in chronic inflammation and fibrosis leading to CKD [16,17] (Figure 1B).

During the past decade, evidence has accumulated supporting the role of AKI as an independent risk factor for CKD progression, including end-stage renal disease (ESRD) [18]. Namely, a significant portion of patients who survive their acute illnesses fail to regain an adequate renal function and progress to CKD and ESRD, requiring maintenance dialysis [19]. In about 8% to 16% of AKI patients, even a single episode of AKI could lead to a progressive loss of renal function and the development of CKD [20].

Due to the high complexity of renal structure and incomplete understanding of the pathophysiology of AKI, regenerative medicine for kidney diseases remains unfeasible. The development of efficient therapeutic strategies that facilitate repair of the damaged renal tissue or prevent progression to CKD is essential to improve short-term and long-term outcomes.

## 3. Extracellular Vesicles: General Features and Biological Relevance

### 3.1. Definition and General Characteristics of EVs

EVs are membrane-enveloped structures, released by virtually all cell types, which transfer biomolecules to other cells. Since EVs are released by both prokaryotic and eukaryotic cells [21] and induce changes in the physiology of the recipient cell, they have been recognized as important mediators of intercellular communications [22]. In addition to their role in physiology, EVs have important roles in various pathophysiological processes [23], which has raised the interest of the scientific community for their exploitation as diagnostic and therapeutic tools for many diseases [24].

According to their biogenesis, there are two main types of EVs: exosomes and microvesicles (MVs) [25]. Even though apoptotic bodies are also considered a type of EVs, it should be noted that their formation occurs only during cell death, and the role of apoptotic bodies in intercellular communication is less studied [26]. Exosomes are formed as intraluminal vesicles by the invagination of the limiting membrane of the early endosome, i.e., the multivesicular body (MVB) [27]. After fusion of the MVB with the plasma membrane, intraluminal vesicles are released in the extracellular space as exosomes. MVs are formed by outward budding of the plasma membrane [28]. Both processes have been described in more detail elsewhere [29,30]. It is generally considered that MVs are larger in size (100–1000 nm) in comparison to exosomes (30–150 nm). Because these two EVs types overlap in size, they cannot be distinguished using the dimensions of the vesicle as a criterion [25,31]. Exosomes and MVs also overlap in their composition; hence, there are still no markers that can distinguish between them. However, there are several markers, including CD63, CD81, and CD9, that are used as general EV markers [32].

EVs carry a variety of molecules including proteins, nucleic acids, lipids, and metabolites and are able to transfer them to recipient cells. The interaction of EVs with recipient cells occurs in one of three ways: (a) interaction of surface molecules on EVs and recipient cells; (b) fusion of EVs with the plasma membrane of target cells; or (c) endocytotic internalization of EVs by target cells [33,34,35]. As one of the fundamental communication tools between cells, EVs are involved in nearly all physiological processes in the organism, including the regulation of renal homeostasis and function [7,8].

### 3.2. EVs as Mediators of Intercellular Communication along the Nephron

All regions of the nephron release EVs which can be traced back to their cell of origin via transcriptomics [36] and proteomics analyses [37].

Several studies have suggested the role of EVs in renal homeostasis. Jella et al. [38] revealed that RPTECs release EVs carrying active GAPDH, which could directly affect the activity of the epithelial sodium channel on distal tubular cells. Gracia et al. [39] demonstrated the presence of miRNAs in urinary EVs, with a role in modulating channel expression and key renal tubular functions in a paracrine manner [39]. Using next generation sequencing, the authors identified a urinary exosomal miRNA signature, where the most abundant miRNAs were miR-10, miR-30, and the let-7 miRNA families, with a potential to regulate the expression levels of potassium channels and amino acid transporters, as well as SGK1 and WNK1 kinases in renal epithelial cells [39]. Communication between different nephron structures, such as RPTECs, the distal tubule, and the collecting duct by means of EVs, was demonstrated by Gildea et al. [40]. Furthermore, EVs can transfer aquaporin 2 (AQP2) between cells of the collecting duct [41] in vitro. The tubular route of this communication is corroborated by the presence of AQP2 containing EVs in the urine [42].

A growing body of evidence supports the involvement of EVs released by podocytes, tubular cells, and endothelial cells in kidney pathophysiology. For instance, podocytes stressed by pressure or hyperglycemia can undergo autophagy and apoptosis and release EVs into the urine [43]. These EVs interact with RPTECs, resulting in activation of the p38-MAPK signaling pathway, and in renal tubulointerstitial fibrosis (TIF) [44]. Injured RPTECs release EVs carrying miR-216a and stimulate nearby RPTECs to undergo epithelial–mesenchymal transition (EMT) and subsequently renal TIF through the PTEN/Akt pathway [45]. RPTECs also shuttle the sonic hedgehog (Shh) signaling protein to fibroblasts, thus mediating their activation [46], while EVs released by hypoxic RPTECs contain TGF-β1 mRNA and also induce fibroblast activation [47]. Glomerular endothelial cells (GEC)- exposed to high glucose released EVs that induced EMT in podocytes and subsequently glomerular barrier dysfunction through the transfer of TGF-β1 mRNA, potentially contributing to diabetic nephropathy [48]. EVs from high glucose-treated GECs also promoted α-smooth muscle actin (α-SMA) expression, proliferation, and extracellular matrix protein overproduction in glomerular mesangial cells (GMCs) via the TGF-β1/Smad3 signaling pathway [49].

Tubular and glomerular cells can also communicate bidirectionally with other cell types involved in kidney injury via EVs. Thus, injured tubular epithelial cells release EVs that can stimulate macrophage activation and infiltration, amplifying inflammation [50]. Importantly, EVs derived from macrophages can regulate podocyte pyroptosis in diabetic nephropathy. A recent study by Ding et al. [51] demonstrated that macrophages stimulated by high glucose secreted EVs carrying miR-21-5p cargo, and their internalization by podocytes increased reactive oxygen species (ROS) production and inflammasome activation.

## 4. EVs Modulate Key Pathophysiological Mechanisms Involved in Organ Injury

Due to their versatile nature and cargo, EVs modulate several pathways involved in the pathophysiology of organ injury, AKI being one of them. Specifically, EVs can influence the onset and propagation of inflammation, renal tubular apoptosis and autophagy, oxidative stress, and cell proliferation [52], thus contributing to the development of organ injury.

### 4.1. Role of EVs in Inflammation and Immune Modulation

During inflammation, both innate immune cells and damaged/infected cells release EVs, and the complex interplay of signals transferred by these EVs regulates the course of inflammation. EVs secreted by damaged/infected cells harbor damage-associated molecular patterns (DAMPs), ligands for pattern recognition receptors (PRRs) in immune cells [53]. Many DAMPs have been identified in EVs, including high mobility group box 1 (HMGB1), histones, heat shock proteins (HSPs), cell-free DNA (cfDNA), and extracellular RNAs (exRNAs) [54]. They are present in EVs either as cargo or as surface-associated molecules. EV-associated DAMPs activate toll-like receptors (TLRs) in macrophages, thus stimulating NF-κB signaling, as well as the release of ROS and inflammatory cytokines [55]. In the healthy kidney, the most numerous resident immune cells are dendritic cells. Upon acute injury, the release of DAMPs or pathogen-associated molecular patterns (PAMPs) provokes an inflammasome response followed by a prompt invasion of neutrophils, monocytes/macrophages, and natural killer (NK) T cells. These cells subsequently interact with adaptive immune effector cells, initiating a cascade that can support the repair (M2 macrophages, Th1, and Treg cells) or further damage to the kidney (plasma B cells and Th2 cells) [56]. During sepsis, DAMPs and PAMPs released in the kidney activate PRRs in immune cells, endothelial cells, and tubular epithelial cells, triggering the secretion of cytokines and promoting inflammation [57]. EVs also contain putative anti-inflammatory miRNAs. Acute stress increases the content of EV-associated DAMPs and reduces intra-exosomal immunoinhibitory miRNAs [58]. Removal of inhibitory signaling, accompanied by increased immune-stimulatory DAMPs, facilitates a rapid stress-evoked inflammatory response.

In response to external stimuli, macrophages, as major modulators of inflammation [59], can polarize towards the pro-inflammatory M1 phenotype or the anti-inflammatory M2 phenotype, secreting secondary cytokines and/or EVs with immunomodulatory capacity [60,61,62,63]. For instance, miRNAs, such as miRNA-19b-3p, which is secreted by tubular epithelial cells during LPS-induced AKI, activates M1 macrophages, subsequently leading to a secretion of MCP-1, IL-1β, IL-6, TNF-α, and iNOS, macrophage infiltration, and tubulointerstitial inflammation. Further down the inflammation cascade, exosomes secreted by macrophages may play a role in AKI [64]. On one hand, M1-released EVs can increase the secretion of pro-inflammatory signals in the trophoblast and the lung in response to infection or acute injury, respectively, as well as in blood vessel endothelial cells during hypertension [65,66,67]. On the other hand, M2-derived EVs can attenuate inflammation by promoting Th2 polarization through CCL1 expressed on EVs’ surface in inflammatory bowel disease [68]. M2-derived EVs also inhibit Akt signaling, downregulate the expression of matrix-metalloproteinases, and reduce the secretion of inflammatory factors [69].

Platelet-derived EVs make up the majority of EVs in the circulation and facilitate inflammation. LPS-stimulated platelets release IL-1β-loaded EVs. Transfer of IL-1β to endothelial cells stimulates the production of endothelial vascular cell adhesion protein 1 (VCAM1), adhesion of monocytes, and inflammation [70,71]. Endothelial cell-derived EVs also attract neutrophils, by transferring kinin B1 receptors and IL-8 [72]. EVs secreted by platelets, leukocytes, and red blood cells also contain complement components [73,74,75,76,77]. Neutrophils readily phagocytose these complement-coated vesicles and initiate an inflammatory cascade.

### 4.2. Role of EVs in Cell Proliferation

EVs directly modulate cell proliferation by transferring factors that contribute to cell proliferation or inhibit cell death. Indirectly, EVs may modulate the microenvironment that supports cell proliferation. In vitro, cancer-derived EVs directly stimulate the proliferative phenotype of recipient cells [78]. This effect is dependent on endocytosis and can be abrogated by dynasore, an inhibitor of dynamin, which is an essential protein for membrane fission during clathrin-mediated endocytosis. Mechanistically, EVs from cancer cells can transfer mRNA encoding hTERT (human telomerase reverse transcriptase) to non-transformed fibroblasts, increasing their proliferation, extending life span, and postponing senescence [79]. Senescent cells may also secrete EphA2-enriched EVs, which further promotes EphA2/ephrin-A1 signaling and cell proliferation [80]. Indirectly, modulation of inflammation by EVs promotes the proliferation of gastric and prostate cancer cells [60]. Cancer-derived EVs affect the proliferation of immune cells in a cell-type specific manner. EVs released from human prostate cancer cells bind to FasL and inhibit T-cell proliferation [81]. In contrast, exosomes released from lymphoma cell lines induce the proliferation and differentiation of B cells [82].

### 4.3. Role of EVs in Oxidative Stress

Oxidative stress arises when ROS production exceeds the ability of the system to eliminate them. In homeostasis, antioxidant enzymes, including superoxide dismutase (SOD), catalase (CAT), and glutathione peroxidase (GPx), eliminate ROS. These enzymes have been identified as components of EVs [83,84,85,86]. EVs are implicated in oxidative stress in a two-fold way: they can either transfer antioxidant enzymes and offer protection, or they can transfer ROS to perpetuate/aggravate oxidative stress, depending on their origin.

EVs also transfer biologically active components, such as apolipoprotein D, to confer protection [87]. During uptake, EVs may alter physiochemical properties of the cell membrane, either by changing the rigidity of the membrane or affecting the exchange of cholesterol, which effectively diminishes the ability of cells, such as neutrophils, to produce ROS [88,89]. In addition to lipids and proteins, oxidative protection may also be conferred by RNAs from EVs, as in the case of mast cells exposed to oxidative stress. EVs released under these conditions attenuate the loss of cell viability [90].

Oxidative stress may directly or indirectly alter EV composition. Loss of putative anti-inflammatory miRNAs may promote inflammation in reaction to stress [58]. The oxidative inactivation of PTP1B in senescent cells increases EphA2 sorting into EVs, to promote cancer cell proliferation through ephrin-A1 binding [80].

### 4.4. EVs and Autophagy

Cells engage in autophagy to degrade obsolete proteins, protein aggregates, and organelles. The components destined for degradation are first enveloped in double membranes (autophagosome) and then fused with lysosomes to be degraded.

Autophagy is closely related to exosome biogenesis through shared molecular machinery or organelles, including autophagy-related genes (ATGs) [91]. ATG5, for instance, regulates autophagosome growth and promotes the fusion of the MVB to the plasma membrane, thus supporting exosome release [92,93,94]. The ATG12-ATG3 complex can either promote autophagosome fusion with lysosomes by catalyzing LC3B conjugation, or the regulation of exosome biogenesis through interaction with ALIX [95]. In both cases, the loss of ATGs leads to reduced exosome release, and the loss of ALIX reduces autophagy flux. This points to reciprocal regulation of exosome biogenesis and autophagy. In mammalian cell lines and mouse models, failure to release exosomes can increase the rate of autophagy [95]. In an opposite scenario, blocking autophagy can rescue or facilitate exosomes secretion [96,97].

### 4.5. Role of EVs in Cell Death

Obsolete and damaged cells need to be removed in a timely and organized manner to maintain tissue homeostasis. Along with microvesicles and small EVs/exosomes produced by all cells, certain EV types are only formed during cell death [98,99,100]. These types include apoptotic extracellular vesicles (ApoEVs) which are comprised of large apoptotic bodies (ApoBDs), or apoptotic microvesicles (ApoMV) and necroptotic exosome-like vesicles. Additionally, membrane fragments of necrotic cells may re-form into EVs extracellularly.

ApoEVs are the best studied class of cell-death-associated EVs. They can have a dual role, acting either as regulators of homeostasis or mediators of inflammation. ApoEVs facilitate the engulfment of apoptotic cells by displaying “find-me or eat-me” signals, such as surface phosphoserine (PS) [101,102]. ApoEVs can also promote tissue repair and proliferation during development via the transfer of growth factors or miRNA-mediated downregulation of adhesion molecules in target cells [103]. They reduce inflammation through the TGF-β signaling pathway [104], but they can also promote inflammation through IL1 and IL6 secretion by transferring nuclear proteins, such as NF-κB, viral material, and/or antibodies [98,105,106,107]. All these findings illustrate that “find-me and eat-me signals” on apoptotic EVs are instrumental in phagocytic clearance, but that the cellular milieu contributes to their cargo and hence to their ability to further induce inflammation.

Ferroptosis, a more recently described form of programmed necrotic cell death, is characterized by iron accumulation and lipid peroxidation [108]. Intracellularly, iron is stored within ferritin [109]. Overloading cells with iron, in the presence of lipid hydroperoxides, leads to a buildup of toxic lipid alkoxy radicals, damage of organelles, and damage to plasma membranes [110]. Interestingly, ferritin is also found in EVs [111]. In epithelial cells, ferroptotic stress, through expression of Prominin2, promotes ferritin transport to multivesicular bodies and secretion through exosomes [112]. In this scenario, exosome production reduces intracellular iron burden and rescues cells from ferroptosis.

## 5. EVs in Modulation of Essential Processes Involved in Tissue Regeneration

During the last decade, EVs have gained significant attention as promising pro-regenerative entities and possible alternatives to cell therapy, due to their involvement in cell differentiation, proliferation, and angiogenesis, as well as the modulation of extracellular matrix turnover during the process of regeneration.

EVs influence the differentiation of cells involved in regeneration, i.e., the repopulation of injured tissue. Specifically, EVs from bone marrow stem cells (BMSCs) induce osteoblast differentiation through miR-196a [113]. Similarly, EVs released from human adipose tissue-derived stem cells (HASCs), during their differentiation to white and beige adipocytes, direct undifferentiated HASCs towards the same type of adipocytes [114]. Furthermore, EVs from differentiating skeletal muscle cells induce myogenesis of stem cells during skeletal muscle regeneration [115]. In addition, EVs from M2 macrophages direct bone mesenchymal stem cells’ differentiation towards osteoblasts by transferring miRNA-5106 [116].

EVs regulate proliferation during tissue regeneration. Specifically, EVs from injured tubular epithelial cells increase fibroblast proliferation to aid tissue repair [47]. In addition, myocyte proliferation in mouse myocardial infarction model is regulated by EVs from cardiac mesenchymal stem cells (MSC) upon Notch signaling [117]. During peripheral nerve repair, EVs from macrophages increase Schwann cell proliferation [118]. Additionally, CD24-positive cells are tubular cells involved in tubular regeneration that localize among proximal tubular cells, but are phenotypically distinct from them [119]. CD24-positive cells might yield scattered tubular cells (STC) that release regenerative EVs and contribute to the repair of injured tubular cells [120].

One of the important steps in tissue regeneration is the activation of fibroblasts and extracellular matrix (ECM) deposition/turnover. EVs exert both negative and positive regulation of fibroblast activation, as summarized by Hohn et al. [121]. Fibroblast-derived EVs participate in positive feedback by activating other fibroblasts [122]. Other cell types also use EVs to activate fibroblasts during regenerative responses, such as injured tubular epithelial cells that release TGF-β1-carrying EVs [47]. During wound healing, circulating monocytes transdifferentiated to “keratinocyte-like cells” release EVs stimulating dermal fibroblasts and increase matrix metalloproteinase-1 (MMP-1) expression [123]. Immune cells can also stimulate MMP production by fibroblasts [124]. Cardiac and dermal fibroblasts may be activated by EVs carrying Wnt proteins [125,126], pointing to an involvement of EVs in spreading profibrotic signals. EVs from mechanically stressed cardiomyocytes inhibit excessive cardiac fibrosis through the delivery of microRNA-378 to cardiac fibroblasts [127]. In addition, EVs from microvascular endothelial cells increase MMP expression and activity in annulus fibrosus cells during the neovascularization of degenerated intervertebral discs [128]. It is worth mentioning that, despite numerous examples of the ability of different stem-cell-derived EVs to induce changes in the activity of recipient cells regarding ECM turnover, information on crosstalk of resident cells in the process of regeneration is relatively scarce.

During regeneration, EVs also target endothelial cells to participate in angiogenesis. Specifically, EVs regulate angiogenesis through modulation of the migration, growth, and differentiation of endothelial cells. For instance, ischemic cardiomyocytes secrete EVs that increase proliferation and the sprouting of endothelial cells, and stimulate the formation of capillary-like structures, thus promoting heart angiogenesis [129]. During bone remodeling, EVs derived from mature osteoblasts promote angiogenesis by inducing the proliferation, migration, and tube formation of endothelial cells through MMP2 and the VEGF/Erk1/2 pathway [130]. In corneal neovascularization, corneal fibroblasts produce EVs carrying MMP14, which regulates endothelial cell proliferation and migration [131].

Given the numerous and diverse cell types involved in the complex process of tissue regeneration, the orchestrated mutual influence between those cells is of critical importance. As is evident from examples above, EVs play a critical role in this process, being the paramount tool for crosstalk between cells throughout all phases of tissue regeneration.

## 6. Extracellular Vesicles and Acute Kidney Injury

Several lines of evidence demonstrate the renoprotective effect of EVs from several sources in ameliorating renal damage in different models of experimental AKI (Table 1). An increasing body of evidence describes the beneficial effect of MSCs-derived EVs from the umbilical cord, bone marrow, and adipose tissue, as well as EVs derived from sources other than MSCs. Here, we describe how EVs from different origin influence renal cells and, consequently, the outcome of kidney injury. Whenever possible, we describe the cellular and signaling mechanisms underlying the beneficial effects of EVs on the kidney (Figure 2).

### 6.1. Bone Marrow MSC-Derived EVs

The protective role of EVs derived from bone marrow mesenchymal stem cells (BM-MSCs) has been demonstrated in various models of AKI. Bruno et al. [132] revealed that human BM-MSC-derived EVs fostered recovery after glycerol-induced AKI in mice and promoted the proliferation and resistance to apoptosis of RPTECs via shuttling specific cellular mRNAs. Furthermore, the administration of a single injection of BM-MSC-EVs promptly after ischemia-reperfusion (I/R) injury-induced AKI fostered recovery and improved kidney function by decreasing inflammation and apoptosis [133,134], as well as promoting cell proliferation [133]. Collino et al. demonstrated an important role for miRNAs transferred by BM-MSC-EVs in recovery after AKI [137]. Specifically, BM-MSC-EVs depleted of Drosha, the RNase enzyme essential for the biogenesis of miRNAs [163], markedly reduced their protective effect, confirming the role of the miRNA cargo in regeneration after AKI [137]. Consistently, miR-199a-3p from BM-MSC-EVs protected against renal I/R injury and apoptosis by modulating Akt and Erk1/2 signaling pathways [138]. BM-MSC-EVs also improved renal function and morphological lesions in cisplatin or gentamicin-induced AKI through proliferative and/or anti-apoptotic mechanisms [135,136]. Importantly, Gregorini et al. [164] demonstrated that the incubation of donated rat kidneys with EVs produced by BM-MSCs immediately prior to transplantation reduced ischemic injury by altering the expression of genes that regulate membrane transport and cell energy metabolism [164].

### 6.2. Umbilical Cord MSC-Derived EVs

The administration of umbilical cord MSC (UC-MSC)-derived EVs was beneficial in different models of AKI. For instance, UC-MSC-EVs injected under the renal capsule protected against cisplatin-induced AKI in rats [139]. Injected EVs ameliorated oxidative stress and apoptosis, as well as fostered cell proliferation in vitro and in vivo [139]. Furthermore, UC-MSC-EVs improved kidney function in vivo and decreased activation of the p38/MAPK pathway and caspase 3 expression in kidney NRK-52E cells in vitro [139]. In this regard, UC-MSC-derived EVs protected from cisplatin-induced kidney injury, inflammation, and apoptosis by modulating autophagy [146]. UC-MSC-derived EVs also showed protection from sepsis-induced AKI by decreasing tubular cell apoptosis and inflammation, along with improving kidney function. Specifically, EVs’ protection was mediated by miR-146b upregulation and NF-κB inhibition in renal tubular cells [148]. The effect of UC-MSC-derived EVs was also evaluated in I/R-induced AKI. Of interest, UC-MSC-EVs protected from kidney injury, apoptosis, and inflammation after I/R in rats by inhibiting CX3CL1 and diminishing macrophage accumulation [140]. In this model, UC-MSC-EVs also had anti-oxidative properties through the suppression of NOX2/gp91 [143]. Recent studies showed that UC-MSC-derived EVs also modulate kidney angiogenesis, as they improve kidney function after unilateral I/R by reducing apoptosis and enhancing proliferation and angiogenesis in a HIF-1α-independent manner [145]. A pro-angiogenic EVs cargo, consisting of VEGF and RNAs, could explain the regenerative potential of EVs [145]. Moreover, UC-MSC-EVs promote recovery from kidney injury by increasing the expression of HGF, accelerating tubular cell dedifferentiation and growth likely through the activation of Erk1/2 signaling by EVs that transferred RNA to injured tubular cells [141]. Human UC-MSC-derived EVs protected against I/R-induced kidney injury by inhibiting mitochondrial fragmentation and reducing apoptosis through miR-30b/c/d [142] and miR-125b-5p/p53 [149], activation of the Nrf2/ARE pathway [144], and OCT-4 recruitment of the Snail pathway in RPTECs [147]. Some of these pathways also increased cell proliferation [147,149]. In vivo imaging showed that EVs preferentially homed in ischemic injured kidneys and located to RPTECs [149].

### 6.3. Placental Tissue MSC-Derived EVs

Liu et al. [152] investigated the role of human placental MSC (hP-MSC)-derived EVs in AKI and demonstrated their enhanced therapeutic effect when incapsulated in a collagen matrix before intrarenal administration. The collagen matrix improved the retention and therapeutic efficacy of hP-MSC-EVs in I/R-induced AKI and facilitated the proliferation and angiogenesis of renal tubular cells, as well as inhibiting apoptosis and endoplasmic reticulum stress [152]. Similarly, Zhang et al. [153] developed RGD (Arg-Gly-Asp) hydrogels to modulate the stability and retention of EVs. Interaction between EVs and hydrogel was mediated by binding of RGD to integrins on the surface of the MSC-EV membrane. EV-RGD hydrogels successfully preserved renal function and decreased morphological damage in I/R-induced AKI by promoting proliferation and autophagy, as well as decreasing apoptosis in tubular epithelial cells. The miRNA let-7a-5p cargo mediated the beneficial effect [153].

### 6.4. Adipose Tissue MSC-Derived EVs

Adipose MSC-derived EVs (AD-MSC-EVs) have also been investigated in the context of regeneration and recovery after AKI. AD-MSC-derived EVs markedly alleviated kidney impairment from I/R injury through decreasing inflammation, apoptosis, and oxidative stress, as well as stimulating renal angiogenesis. Combined therapy with AD-MSC-derived EVs and adipose-derived MSCs led to an even more superior effect than that of EVs alone [150]. In sepsis-induced AKI, AD-MSC-derived EVs inhibited kidney inflammation and apoptosis through the SIRT1 signaling pathway [151].

### 6.5. EVs Derived from Other Sources

Emerging evidence supports the beneficial role of EVs derived from kidney or liver resident MSCs, tubular epithelial cells, macrophages, or endothelial progenitor cells (EPCs) in experimental AKI.

Kidney-derived MSC-EVs improved renal function and morphological changes in I/R-induced AKI by stimulating the proliferation of peritubular capillary endothelial cells and ameliorating peritubular microvascular rarefaction through the horizontal transfer of mRNA with proangiogenic properties [154]. Glomerular-MSC-EVs protected against I/R-induced AKI through a transfer of miRNAs cargo and activation of cell proliferation [155]. Tubular epithelial cells derived-EVs containing CD26 also alleviated I/R-induced AKI by decreasing inflammation and stimulating cell proliferation via decreasing p53 and p21 [158].

Interestingly, human urine stem cell (USC)-derived EVs protected against I/R-induced AKI and inhibited inflammation and apoptosis by transferring miR-146a-5p and subsequently inhibiting NF-κB activation [159]. Similarly, liver MSC-derived EVs improved glycerol-induced AKI in mice [156]. Kidney-derived EVs from normal urine (uEVs) improved the recovery from glycerol-induced AKI by stimulating tubular cell proliferation, restoring endogenous Klotho levels, and decreasing inflammation through the transfer of miRNAs cargo and Klotho to resident kidney cells [157].

EPC-secreted EVs promoted kidney regeneration in I/R-induced AKI through shuttling miRNA cargo to resident tubular epithelial cells that activate regenerative programs [161]. Moreover, EPC-secreted EVs carrying miR-93-5p protected against sepsis-induced AKI by attenuating vascular leakage, inflammation, and apoptosis through the regulation of the H3K27me3/TNF-α axis [162].

Finally, loading EVs derived from macrophages with IL-10 resulted in enhanced stability of vesicles and the effective targeting of renal tubular cells and macrophages within injured kidneys to alleviate kidney injury, inhibit inflammation, and promote mitophagy through the suppression of mTOR signaling [160].

## 7. Extracellular Vesicles: A Novel Therapeutic Avenue for Kidney Injury

### 7.1. Regenerative Medicine Paradigm Shift from Stem Cells to Stem Cell EVs

Application of stem cell (mesenchymal, embryonic, or induced-pluripotent) therapy to replace damaged cells is the most common biomedical approach in the field of regenerative medicine [2]. Despite their great promise, stem cell therapies are only approved by the FDA for clinical use in graft versus host disease (GvHD) [3]. Therapeutic application is hindered by safety concerns that involve transferring mutated or damaged DNA or the trapping of cells in pulmonary capillaries after intravenous injections due to a large cell size [165,166]. Recently, many of the stem cell therapeutic effects have been attributed to the secretion of paracrine factors, including EVs.

EVs have great potential in replacing cellular therapies and overcoming stem cell therapy challenges [5]. They are well-tolerated by the immune system; they do not elicit adverse immunological responses [167]; and they are not toxic [7,167]. EVs are considered superior to cellular therapy because they cannot proliferate [8,168,169,170]. Furthermore, EVs are capable of crossing blood-tissue barriers [171] and can be loaded with therapeutic molecules [172]. EVs can be formulated as an “off-the-shelf” product, with simple storage and transport, and without the need to monitor differentiation and viability, as is the case with therapeutic cells/MSCs [5].

Numerous studies have shown the protective effects of EVs from different cellular origins in virtually all disease models investigated. Specifically, beneficial effects were shown in GvHD [173], myocardial ischemia [174], vascular regeneration [175], bone regeneration [176], cartilage regeneration [177], lung diseases [178,179,180], liver diseases [181], CKD [182], and AKI, as summarized in the previous section. Currently, MSCs-derived EVs are being tested in clinical trials as therapeutic agents for, among other conditions, COVID-19 and related conditions (ClinicalTrials.gov: NCT04493242, NCT05125562, NCT05116761), dystrophic epidermolysis bullosa (ClinicalTrials.gov: NCT04173650), burn wounds (ClinicalTrials.gov: NCT05078385), refractory Crohn’s disease (ClinicalTrials.gov: NCT05130983), venous trophic lesions (ClinicalTrials.gov: NCT04652531), acute ischemic stroke (ClinicalTrials.gov: NCT03384433), and transplant rejection (ClinicalTrials.gov: NCT05215288).

Given all of the available information on the beneficial roles of EVs in different preclinical models and clinical studies, as well as their versatile roles in different models of experimental AKI, it is reasonable to propose the use of EVs as a novel strategy for the treatment of renal damage post-AKI. Although the production of clinical-grade exosomes has recently been reported [183], before the wider use of EVs as therapeutics there are still many issues to be resolved, as summarized by Nagelkerke et al. [6]. Some of these issues include the definition of the optimal dose, the route of delivery, and the exclusion of off-target/side effects, as well as the upscaling of EVs production.

### 7.2. Pharmacological Application of EVs and Specificities of Therapeutic Application in Kidney Disease

The optimal dose of EVs depends on the quantity of the active substance and its ability to reach the target tissues/cells [184]. There are no standardized criteria for EVs dosing, and preclinical studies use protein concentration and various particle-counting approaches to describe the applied dose. Additionally, the mechanisms of EVs’ action, i.e., the action of the active molecules, or more likely the combination of molecules in natural (not modified) EVs, has not been fully elucidated. Considering that cells produce a heterogenous population of EVs, it is conceivable that not all of the therapeutic preparations of EVs carry active molecules or the same amount of them [185].

In addition to the tissue source of MSCs used to produce therapeutic EVs (Table 1), other parameters, including the dosing regimen and the time of application, should be considered. Regarding the regimen of EVs’ application in different models of experimental kidney disease, EVs were most commonly given in a single dose in experimental I/R-induced AKI [133,134,139,141,142]. However, multiple doses are reported in different models of AKI and CKD (cisplatin or gentamicin, diabetes, aristolochic acid nephropathy) [132,136,186,187]. The time between the onset of injury to EVs treatment can vary from several minutes (I/R injury) to several days (I/R injury, gentamicin, or cisplatin) [187]. These studies also show that the therapeutic EVs dosage varies greatly, regardless of the dose being established on protein content (15–200 µg) or particle count (10^7^–10^10^) [187]. Generally, therapeutic EVs dosage depends on the source of MSCs, the disease model, and the likely technical aspects of EVs processing.

Biodistribution studies show that therapeutic EVs accumulate in clearance organs, mainly the liver, lung, and spleen, shortly after systemic injection [188,189]. When EVs doses are increased, accumulation occurs in the kidney as well [188,190,191], despite the short circulation time of EVs [192,193]. This may prove to be an advantage in the treatment of kidney conditions, as it would override the usual need for improved targeting. The circulation time for EVs can be increased by incorporating “don’t eat me” signals in EVs or incorporating EVs in biomaterials intended for their slow release [194,195]. However, since EVs can trigger different signaling pathways in the cell and interact with different cell types, they can cause undesirable off-target side effects, especially during long-term application, emphasizing the need for controlled targeting [6].

For EVs-based therapies to enter routine clinical use, it will be necessary both to elucidate their mechanism of action and enhance their targeting. Enhancement of the targeting properties of EVs can be achieved by adding/enriching the EVs’ surfaces with molecules that recognize specific cells (antibodies, ligands, receptors, and sugar moieties) or by incorporating molecules on their surfaces that are activated at target sites (i.e., in tumor environment) [196].

Production of EVs in sufficient quantities and clinical-grade quality requires up-scaling and standardizing both cell culture and EVs isolation/purification techniques [6,197,198,199]. However, upscaling cell culture usually implies use of methods different from those in research laboratories, some of which can lead to changes in the composition of the media, occurrences of shear stress, decreases in the availability of nutrients, or increases in the levels of toxic metabolites [200], all of which can change in the composition of EVs. This in turn points out the need for standardization of quality control methods, i.e., EVs characterization, yield measurement, and potency determination, to ensure uniformity and safety in the final preparation.

To solve some of these issues, strategies for the modification of EVs or the production of “artificial EVs” are also being developed. Thus, cells can be genetically modified to express molecules of interest (i.e., miRNA) [196], or can be grown with molecules intended to be loaded in EVs [201]. Alternatively, EVs can be loaded with drugs/biomolecules by electroporation or other methods [202]. However, using either natural or modified EVs implies, as Silva et al. [203] stated, using “semi-black boxes” with uninvestigated effects of most molecules in its composition. In another approach, “artificial EVs” offer easier and low-cost production, as well as better quality control [204]. The goal is to make functionalized liposomes, loaded with active molecule(s) and decorated with targeting molecules. However, this field is still in early stages.

In addition to the issues mentioned above, in designing EVs-based therapeutics for the treatment of kidney disorders, it should be kept in mind that EVs have larger diameters than those of the pores of the slit diaphragm and cannot pass a preserved glomerular filtration barrier (GFB) [205]. Thus, it would be of interest to clarify whether EVs may reach kidney cells from the urinary or interstitial side or whether this occurs both under physiological or pathological conditions, as nephron cells are polarized under physiological conditions but may lose both polarity and markers under injury conditions. This would be especially relevant if molecules on the cell surface of kidney epithelial cells are to be used as EVs anchors. Despite these considerations, there are indications that EVs administered intravenously do reach renal cells [206], possibly due to the disrupted tissue architecture and general increase of non-specific phagocytosis in injured tissues [6], in addition to the non-specific accumulation mentioned earlier. Although the optimal administration route for therapeutic EVs has still not been elucidated, and in many cases their mechanism of action is vague, numerous examples of the beneficial effects of EVs (mainly MSC-derived ones) show their great potential to be used as a basis for the development of a novel therapeutic for AKI. Some of the promising approaches may encompass MSCs EVs modified to enhance their targeting to the kidney or the formulation of “artificial” vesicles that would carry miRNA as in natural EVs, along with targeting moieties. In addition, strategies that could facilitate the delivery of active molecules found in EVs through the GFB would be of great value. One such approach, yet to be explored in detail, is the delivery of nanoparticles able to cross the GFB [207].

## 8. Conclusions

Extracellular vesicles are membrane-enveloped nanoparticles with essential roles in cell-to-cell communication. Owing to their ability to cross blood-tissue barriers transferring active biomolecular cargo, EVs are considered key players in diverse physiological and pathological conditions. EVs are critical mediators of intercellular communication along the nephron and can modulate important pathophysiological mechanisms involved in AKI, such as inflammation, cell proliferation, apoptosis, autophagy, and oxidative stress. Furthermore, therapeutic EVs from diverse sources have attracted significant attention as being important tools for kidney repair and regeneration through regulation of differentiation, angiogenesis, and ECM turnover.

One plausible target of therapeutic EVs in AKI could be the RPTEC (Figure 3). RPTECs are exceptionally vulnerable to different types of injury due to their high metabolic demands. After injury, RPTECs display a broad repertoire of molecular, morphological and functional alterations that subsequently lead to tubule regeneration or, in the case of sustained injury or failed repair, to a degeneration of the tubular structure. EVs released from various sources, including MSCs, the kidney, or the liver, as well as from EPCs, have emerged as a powerful cell-free therapy in different models of experimental AKI. Active biomolecules carried by MSC-derived EVs may alleviate kidney damage by decreasing apoptosis and oxidative stress, and by promoting the proliferation and autophagy of RPTECs. Shuttling different molecules, MSC-secreted EVs can also target peritubular capillary endothelial cells, decreasing microvascular rarefaction and promoting angiogenesis, and/or macrophages, stimulating M2 macrophage polarization.

Considering the clear evidence on the protective and regenerative role of EVs in various preclinical and clinical studies, it is feasible to suggest the use EVs as a novel avenue for renal repair after AKI.

Some promising approaches may involve MSC-EVs adapted to enhance their targeting and delivery towards kidneys, or a design of “artificial” vesicles carrying active biomolecules as in natural EVs, along with targeting moieties. The success of these novel therapeutics depends on overcoming technical and production issues, which, given rapid development of EVs field, might be accomplished in the near future.

## Figures and Tables

**Figure 1 ijms-23-03792-f001:**
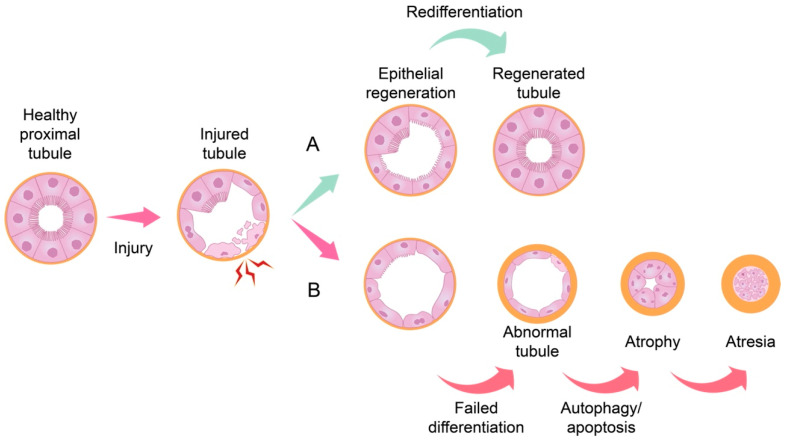
Schematic representation of renal tubule repair after injury. After initial tubular injury, remaining healthy tubular epithelial cells undergo dedifferentiation and proliferate to repair damaged tubules. (**A**) Normal repair can result in full restoration of tubular epithelial structure and function. However, (**B**) continuous injury or impaired differentiation may lead to abnormal tubules, thickening of the basement membrane and subsequently to autophagy and apoptosis. Defective regeneration and repair will eventually result in the development of chronic inflammation and fibrosis leading to chronic kidney disease.

**Figure 2 ijms-23-03792-f002:**
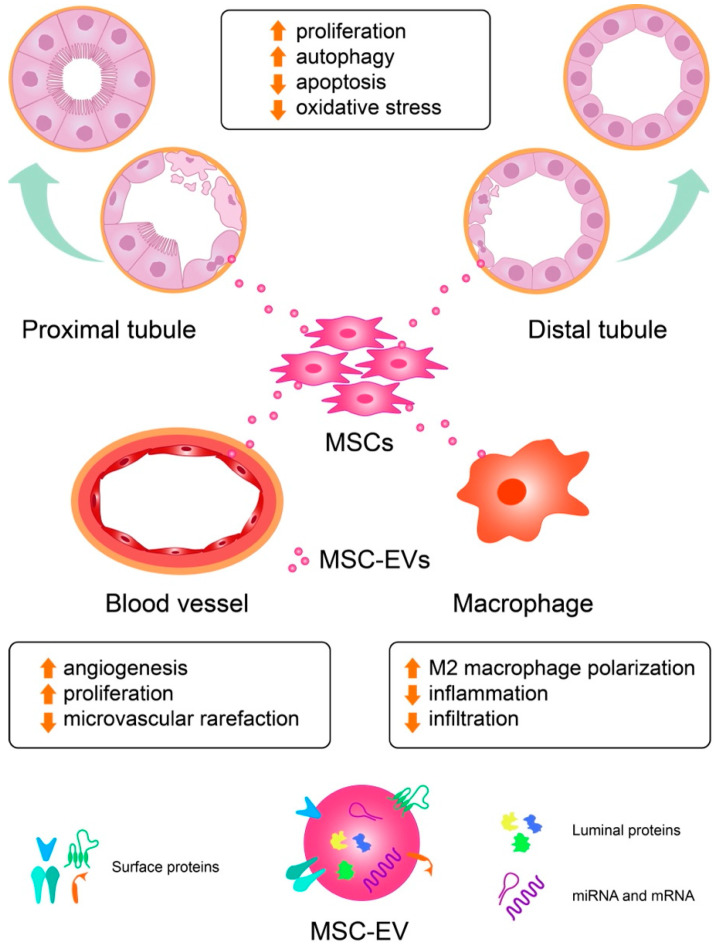
Extracellular vesicles are renoprotective by modulating key pathways implicated in the pathophysiology of kidney injury. Administration of MSC-derived EVs protects from kidney injury by decreasing apoptosis and oxidative stress and stimulating the proliferation and autophagy of proximal and distal tubular cells. Furthermore, MSC-derived EVs can promote angiogenesis and decrease microvascular rarefaction, as well as stimulating M2 macrophage polarization and decreasing macrophage infiltration.

**Figure 3 ijms-23-03792-f003:**
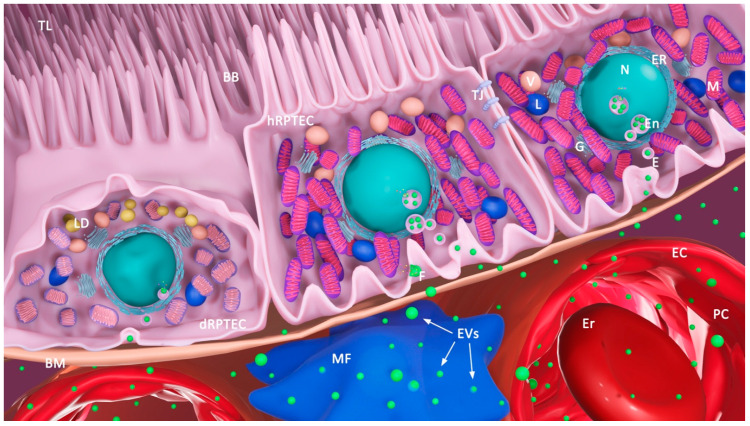
Renal proximal tubular epithelial cell as a target of therapeutic EVs. Therapeutic EVs can deliver their cargo (proteins, RNA, and other biomolecules) to the RPTEC either by fusion (F) with its plasma membrane and release of the cargo into the RPTEC´s cytoplasm or endocytotic uptake (E) of EVs by the RPTEC. BB, brush border; TL, tubular lumen; BM, basal membrane; En, endosome; TJ, tight junction; dRPTEC, diseased RPTEC; hRPTEC, healthy RPTEC; M, mitochondria; LD, lipid droplet; V, vacuole; L, lysosome; G, Golgi apparatus; ER, endoplasmic reticulum; N, nucleus; EV, extracellular vesicle; E, endocytosis; F, fusion of EVs with plasma membrane of RPTEC; MF macrophage; PC, peritubular capillary; EC, endothelial cell; Er, erythrocyte; EVs released from RPTEC are not shown.

**Table 1 ijms-23-03792-t001:** Application of EVs as therapeutic agents in acute kidney injury.

EV Source	AKI Model	EV Cargo	Signaling Pathway	Mechanism	Administration	References
BM-MSCs	Glycerol	mRNA	n/i	Proliferation, Apoptosis	Intravenous	[132]
	I/R injury	RNA	n/i	Proliferation, Apoptosis	Intravenous	[133]
	I/R injury	CCR2	NF-κB p65	Inflammation	Intravenous	[134]
	Cisplatin	n/i	n/i	Proliferation, Apoptosis	Intravenous	[135]
	Gentamicin	RNA	n/i	Proliferation, Apoptosis	Intravenous	[136]
	Glycerol	miRNA	n/i	Inflammation	Intravenous	[137]
	I/R injury	miR-199a-3p	Akt, Erk1/2	Apoptosis	Intravenous	[138]
UC-MSCs	Cisplatin	n/i	p38/MAPK, Erk1/2	Oxidative stress, Apoptosis, Proliferation	Renal capsule	[139]
	I/R injury	n/i	CX3CL1	Apoptosis, Inflammation	Intravenous	[140]
	I/R injury	HGF/RNA	Erk1/2	Proliferation, Apoptosis	Intravenous	[141]
	I/R injury	miR-30b/c/d	n/i	Apoptosis	Intravenous	[142]
	I/R injury	n/i	NOX2/gp91	Oxidative stress, Apoptosis, Proliferation	Intravenous	[143]
	I/R injury	n/i	Nrf2/ARE	Oxidative stress, Apoptosis	Intravenous	[144]
	I/R injury	VEGF, RNAs	n/i	Apoptosis, Proliferation, Angiogenesis	Intravenous	[145]
	Cisplatin	n/i	n/i	Inflammation, Apoptosis, Autophagy	Renal capsule	[146]
	I/R injury	Oct-4	Snail	Apoptosis, Proliferation	Intravenous	[147]
	Sepsis	miR-146b	NF-κB	Apoptosis, Inflammation	Intravenous	[148]
	I/R injury	miR125b-5p	p53	Apoptosis, Proliferation	Intravenous	[149]
AD-MSCs	I/R injury	n/i	n/s	Inflammation, Apoptosis, Oxidative stress, Angiogenesis	Intravenous	[150]
	Sepsis	n/s	SIRT1	Apoptosis, Inflammation	Intravenous	[151]
P-MSCs	I/R injury	n/i	n/i	Proliferation, Angiogenesis, Apoptosis	Intrarenal	[152]
	I/R injury	Let-7a-5p	n/i	Proliferation, Apoptosis, Autophagy	Intrarenal	[153]
K-MSCs	I/R injury	mRNA	n/i	Proliferation, Angiogenesis	Intravenous	[154]
	I/R injury	miRNAs	n/i	Proliferation	Intravenous	[155]
L-MSCs	Glycerol	n/i	n/i	Proliferation, Apoptosis	Intravenous	[156]
u-EVs	Glycerol	miRNA, Klotho	n/i	Proliferation, Inflammation	Intravenous	[157]
TECs	I/R injury	CD26	p53, p21	Proliferation, Inflammation	Intravenous	[158]
USCs	I/R injury	miR-146a-5p	NF-κB	Apoptosis, Inflammation	Intravenous	[159]
Mac	I/R injury	IL-10	mTOR	Inflammation, Autophagy	Intravenous	[160]
EPCs	I/R injury	miRNAs	n/i	Proliferation, Apoptosis	Intravenous	[161]
	Sepsis	miR-93-5p	H3K27me3/TNF-α	Inflammation, Apoptosis	Intravenous	[162]

EVs, Extracellular vesicles; AKI, acute kidney injury; MSCs, mesenchymal stem cells; BM-MSCs, bone marrow MSCs; UC-MSCs, umbilical cord MSCs; AD-MSCs, adipose tissue MSCs; P-MSCs, placental MSCs; K-MSCs, kidney resident MSCs; L-MSCs, liver resident MSCs; uEVs, renal derived EVs isolated from urine; TECs, tubular epithelial cells; USCs, urine-derived stem cells; Mac, macrophages; EPCs, endothelial progenitor cells; MAPK, mitogen-activated protein kinase; ARE, antioxidant response element; VEGF, vascular endothelial growth factor; mTOR, mammalian target of rapamycin; I/R, ischemia-reperfusion; n/i, not investigated; n/s, not specified.

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
