# Peer review of "Extracellular Vesicles and Acute Kidney Injury: Potential Therapeutic Avenue for Renal Repair and Regeneration"

_ijms, 2022, doi:10.3390/ijms23073792_

Round 1

Reviewer 1 Report

This manuscript is generally written well and is of scientific interest. 

In this manuscript, roles of EVs in pathophysiological mechanisms related to AKI and renoprotective effects of EVs from different origins in AKI were summarized with explaining known EVs function in AKI and potential clinical move of EVs at AKI. 

I think that this theme are suitable to this journal’s policy. 

However there are some requirements to add to relate with AKI and immune, and show interactions between EVs move and immune at occurrence of AKI. 

In Item 4, “EVs modulate key pathophysiological mechanisms involved in organ injury”, generally because we are able to image immune response’s change at AKI occurrence, please add more relationship among specific immune response in AKI and EVs move considering with their mechanisms. EVs and the other must modulate specific immune responds at AKI occurrence and many signal in immune cells. 

Reviewer 2 Report

The manuscript presents an interesting,  comprehensible, well written and useful review.

Remarks:

Lines 259-263: some of the enzymes mentioned are typical for plant rather than animals cells and are not mentioned in the publications cited, except for catalase ([80, 81]). In turn, glutathione peroxidase is not mentioned.

Figure 3: Wouldn’t it be better to indicate  e.g. “Therapeutic EVs (green) can deliver…” The inscription “EV” may be not well readable at first sight. Or you may  direct arrows from “EV” to some of the nearest vesicles.
